# Immune Response to COVID-19 mRNA Vaccine—A Pilot Study

**DOI:** 10.3390/vaccines9050488

**Published:** 2021-05-11

**Authors:** Andrzej Lange, Agata Borowik, Jolanta Bocheńska, Joanna Rossowska, Emilia Jaskuła

**Affiliations:** 1Hirszfeld Institute of Immunology and Experimental Therapy, Polish Academy of Sciences, 53-114 Wroclaw, Poland; joanna.rossowska@hirszfeld.pl (J.R.); emilia.jaskula@hirszfeld.pl (E.J.); 2Lower Silesian Center for Cellular Transplantation with National Bone Marrow Donor Registry, 53-439 Wroclaw, Poland; borowik@dctk.wroc.pl (A.B.); bochenska@dctk.wroc.pl (J.B.)

**Keywords:** COVID-19, SARS-COV-2, mRNA vaccine, antibodies response, IFN gamma blood cells release

## Abstract

Twenty individuals (17 females, 3 males, aged 31–65 years (range), median: 46) who received both doses of the BioNTech Pfizer mRNA vaccine were examined (11 to 31 days, median: 25) after the second dose for the presence of antibodies against peptides of SARS-COV-2 and some of MERS-CoV, SARS-CoV1, HCov229E, and HCoVNL63. Clinical evaluation revealed that six people had COVID-19 in the past. We found that: (i) Six people claimed the presence of unwanted effects of vaccination, which were more frequent in those with a history of COVID-19 (4 out of 6 vs. 2 out of 14, *p* = 0.037); (ii) All individuals independent of the past history of COVID-19 responded equally well in IgG but those who experienced the disease tended to do better in IgA class (729.04 vs. 529.78 U/mL, *p* = 0.079); (iii) All those who had experienced the disease had IgG antibodies against nucleocapsid antigens but also 5 out of 14 who had not had the disease (6/6 vs. 5/14, *p* = 0.014); (iv) Anti S2 antibodies were present in the patients having COVID-19 in the past but also were found in those who had not had the disease (6/6 vs. 8/14, *p* = 0.144); (v) All vaccinated people were highly positive in the IGRA and the level of released IFN gamma was correlated with the numbers of HLADR positive lymphocytes in the blood (*R* = 0.5766, *p* = 0.008).

## 1. Introduction

COVID-19 is spreading throughout the globe with an unprecedented death toll and social and economic disaster. The experience of the last year highlights that the social measures are not adequately effective. It was understandable from the very beginning that vaccination against COVID-19 can save the lives of millions and may open the route back to normality. Several scientific institutions and companies made an enormous effort to develop the vaccine [1]. The most promising approach is based on an mRNA platform. It offers speed of production, which, in addition, is flexible enough to keep pace with the emerging new variants, which may escape from the primary vaccine evoked immune response [2].

The mRNA vaccine developed either by Pfizer BioNTech (Pfizer Manufacturing Belgium NV, Puurs, Belgium, BioNTech Manufacturing GmbH, Mainz, Germany) or Moderna (Moderna Biotech Spain, S.L., Madrid, Spain) made the vaccination roll out possible quite recently [3]. The introduction of the vaccine was unprecedently fast, a consequence of outstanding scientific achievements. The experience of the countries leading in the vaccination process has already shown a decline in the pace of virus transmission [4], and the research laboratories have shown a good response to vaccination in activating Th1 lymphocyte and antibody production [5]. This information is extremely important for now, but nevertheless, the fast timeline in rolling out the vaccination left the question of how long the post-vaccination immunity lasts still unanswered. The present global effort is to vaccinate as many people as possible, hence, groups of people, beginning with those prioritized, constitute waves of thousands, making the evaluation of the possible previous exposures to COVID-19 unfeasible. The estimated frequency of individuals who have already contracted SARS-COV-2 is not well established, mostly due to the scarce symptomatology of COVID-19 among the younger population [6]. Hence, in the vaccinated population, there are two subgroups with respect to the past history of COVID-19. It makes it possible to compare the profile of SARS-COV-2 antibodies which arose solely after vaccination with that in people who have already had COVID-19 prior to vaccination. These data may help in the future development of vaccines.

This study demonstrated that the presence of receptor-binding domain (RBD) antibodies is attributable to vaccination, whereas N and S2 antibodies bear witness to past positive SARS-COV-2 history.

## 2. Materials and Methods

All employees in the social care delivery institution were invited for volunteer participation in the study on the level of immunity against SARS-COV-2 after the second dose of the Pfizer-BioNTech mRNA vaccine. For the pilot study, 20 people were invited: 17 females, 3 males, aged 31–65 years (range), median: 46 years (including five medical professionals not involved directly in COVID-19 health care delivery).

All participants provided signed consent to be studied, being informed and examined by one of us (A.L.) according to the standard protocols for COVID-19 vaccination. All side effects, if any, were recorded. They were from 11 to 31 days (median 25 days) after the second dose.

On the day of examination, blood was collected to ensure laboratory work as described below.

All the patients were clinically evaluated, which resulted in stratification of the whole group into two subgroups composed of individuals either having or lacking a positive history of COVID-19. In the latter group, all except one (a spouse of a COVID-19 positive case) were positive in the genetic test for SARS-COV-2. In the next stratification approach, the whole group was divided regarding the presence or absence of side effects (clinically assessed as muscle and joint pain, dyspnoea, weakness, fever, enlarged lymph nodes, headache, eye pain).

In the studied cohort, six people had a previous history of COVID-19 (in four to five months prior to vaccination), and six vaccinated people had post-vaccination side effect symptoms. Both groups overlapped as four individuals had both COVD-19 in the past and post-vaccination side effects.

### 2.1. Microblot-Array for the Detection of SARS-COV-2 Antibodies (Cat. No: CoVGMA96, CoVMMA96 and CoVAMA96, Test Line Clinical Diagnostics s.r.o., Brno, Czech Republic)

In this method, specific recombinant proteins/antigens spotted onto a nitrocellulose membrane are exposed to the tested sera and the reaction is developed using anti-human IgG or IgA or IgM antibodies labeled with alkaline phosphatase. SARS-COV-2 markers include those of N, RBD, Spike S2, E, and in addition, each set of the peptides targeted by tested sera included MERS-CoV S1, SARS-CoV1 N, HCov229E: N, HCoVNL63: N as well as ACE2 and PLPro. The reference values were standardized against 20/136 First WHO International Standard Anti-SARS-COV-2 Immunoglobulin and 20/162 NIBSC Anti-SARS-COV-2 Antibody Diagnostic Calibrant standards. The staining and reading procedures were performed according to the manufacturer’s guidelines, including the threshold of positivity set at 210 U/mL.

### 2.2. Interferon Gamma Release Assay

The test was performed using SARS-COV-2 IGRA (ca no.: ET 2606-300) and the reading of the levels of interferon gamma in the cultured medium using the ELISA (ca no.: EQ 6841–9601), both provided by Euroimmun Medizinische Labordiagnostika, Luebeck, Germany). The tubes were coated with antigens of S1 domain. The tubes coated either with S1 domain or mitogen were provided by the manufacturer. Mitogen coated tubes and the plain ones served as controls. The tested blood was incubated for 24 h at 37 °C, then the blood was centrifuged (12,000× *g*, 10 min) and plasma was collected for IFN gamma content measurements.

### 2.3. Peripheral Blood Lymphocytes Flow Cytometric Analysis

Staining of unfractionated fresh blood was performed according to the lyse and stain approach using monoclonal antibodies: CD45 FITC (clone 2D1), CD4 PE/CF594 (clone RPA-T4, BD Biosciences, San Jose, CA, USA), CD3 PE-Cy7 (clone UCHT1), CD8 BV510 (clone SK1), CD16 PE (clone B73.1), HLADR BV786 (clone L243, BioLegend, San Diego, CA, USA) and the results read in the Fortessa flow cytometer (BD Biosciences, San Jose, CA, USA). Living cells (stained with LIVE/DEAD Fixable Dead Cell Stain Kit with BV421 fluorochrome, Life Technologies Carlsbad, CA, USA) were evaluated. The gating was done using both CD45 and side scatter signals. NovoExpress Software (Agilent Santa Clara, CA, USA) was used for subpopulation analysis.

### 2.4. Statistical Analysis

Statistical analysis was conducted using Statistica v.12 (Stat-Soft Inc., Tulsa, OK, USA). The associations between two variables were tested by Fisher’s exact test for categorical variables and the Mann–Whitney U test (UMW) or median test was used for categorical and continuous variables. Additionally, correlations were calculated using Spearman’s rank correlation test. Differences between samples were considered significant at *p* < 0.05.

## 3. Results

The post-vaccination side effects were found more frequently in the people having than in those lacking COVID-19 history (4/6 vs. 2/14, *p* = 0.037). The side effects were seen from two to seven (median: two) days after the second dose, they were rather mild but raised awareness. Symptoms included: Muscle and joint pain, dyspnoea, weakness, fever, headache, eye pain. Only one person claimed the presence of moderate side-effects, which included the lymph node enlargement under the arm and in the supraclavicular area, that lasted about 10 days. The response to immunization was not different in cases having or lacking vaccination side-effects.

IgG antibodies directed against the receptor-binding domain (RBD) at a level far exceeding the threshold of positivity, established according to the WHO standard and recommendation, were found in all vaccinated people. Of note, the level of RBD antibodies was similar in all people irrespective of having COVID-19 in the past (662.69 ± 43.06 vs. 615.68 ± 13.29 U/mL, *n* = 20, *p* = ns (not statistically significant), Figure 1). The individuals who had experienced this disease had all IgG antibodies against nucleocapsid antigens as compared to those lacking symptoms of overt COVID-19 in the past (6/6 vs. 5/14, *p* = 0.014, Figure 1A and Figure 2).

IgG S2 specific antibodies were seen in all who experienced COVID-19, but they were also seen, however, less frequently in those who denied having the disease in the past (6/6 vs. 8/14, *p* = 0.144, Figure 2), nevertheless, individuals with positive COVID-19 history had antibodies at higher levels than those denying COVID-19 any time before (880.69 ± 33.67, *n* = 6 vs. 431.15 ± 92.89 U/mL, *n* = 8, *p* = 0.009, Figure 1A).

IgA class antibodies at levels exceeding the threshold of positivity were detected in 17 out of 20 vaccinated people. All these positive cases had IgA anti-RBD antibodies. Notably, the level of these antibodies was higher in the individuals with positive than in those with negative COVID 19 history (729.04 vs. 529.78 vs. U/mL, *p* = 0.079). When anti-S2 antibodies were evaluated, COVID-19 positive people were the only ones with these antibodies (3/6 vs. 0/14, *p* = 0.018). Similarly, IgA N antibodies were present only in one person having positive COVID-19 history.

IgM class antibodies were predominantly of RBD specificity, being present in the same proportion in the patients having or lacking COVID-19 positive history in 10 out of 20 cases (Figure 2). IgM S2 and N antibodies were seen in two and one case out of 20 individuals examined, respectively. Their presence was not associated with the past history of COVID-19. IgM response against RBD was rather higher in the individuals who received vaccines but denied the disease as compared to those who were vaccinated and previously experienced COVID-19. It is our belief that it reflects a primary response to Pfizer BioNTech vaccine in individuals not contracting SARS-CoV2- before.

None of the individuals had antibodies detected against epitopes of common coronaviruses, but one reacted with MERS-CoV S1 protein in IgA class.

IFN gamma release assays (IGRA test) documented an abundant output of IFN gamma in response to the S1 peptides. All cases had IFN gamma in the supernatants taken from the culture tubes in the range 888–5923 mUI/mL (median 3393 mUI/mL, Figure 1B), far exceeding the reading of the positive control standardized against the NIBSC 87/576 reference. The supernatants in the control tubes lacking SARSCoV2 peptides contained IFN gamma in the range 0–382 mUI/mL (median: 0 mUI/mL).

Blood was taken for the flow cytometry analysis, which was a part of routine laboratory screening. Lymphocyte count was 1.1 to 3.9 × 10^3^ cells/μL, median 2.3 × 10^3^ cells/μL. There were no differences in lymphocyte count between the groups. The staining included CD3, CD4, CD8, CD16, and HLADR antigens.

Among the individuals having or denying COVID-19 as well as having or lacking side effects, no significant differences were found in the metrics of the readings (Figure 3). However, when the whole cohort was evaluated, independently on the presence or absence of either COVID-19 positive history or side effects, the IGRA response (after S1 domain stimulation) was correlated with the numbers of HLADR + T lymphocytes in the fresh blood (*R* = 0.5766, *p* = 0.008, Figure 1C).

People who experienced post-vaccination side effects did not differ in blood lymphocyte profile (not shown) and IFN gamma release assay, Figure 1B, from those lacking any symptoms after immunization.

## 4. Discussion

In this study, we observed that at 25 days (median) after the second dose of the Pfizer BioNTech mRNA vaccine, all immunized individuals developed anti-RBD antibodies at an average level of over 600 U/mL. The effect of the past history of COVID-19 was negligible in IgG class antibodies but was observed when IgA antibodies were evaluated. This may suggest that COVID-19 patients have a good memory of IgA class response to S1 SARS-COV-2 peptides. Indeed, the recent study of Sterlin et al. [7] documents the prevalence of an IgA response at an early stage of COVID-19 with a high representation of IgA-positive plasma blast at the mucosal barrier.

S2 antibody levels measured after the vaccination were rather higher in the individuals having than in those lacking COVID-19 in the past. A similar association was not seen when anti-RBD antibody levels were evaluated (Figure 1A). In addition, all COVID-19 patients had S2 antibodies but not all denied the disease (Figure 2). It suggests the environmental exposure may play a role in shaping S2 antibody response to vaccination [8]. Likely including other viruses as S2 peptides are similar in structure among coronaviruses This observation warrants further study on a large cohort. The presence of N antibodies is strongly suggestive of the previous positive COVID-19 history (Figure 1A and Figure 2).

All vaccinated people were highly positive in the IGRA assay, which additionally corroborates the efficacy of the immunization and indicates the crucial T cell engagement in the immune response to vaccination, similarly, as it is seen in COVID-19 cases irrespective of the severity of the disease [9].

The limitation of this study is due to rather a small number of people investigated, but the results show the area of interest for planning large-scale investigations.

In summing up, the response to SARS-COV-2 spike glycoprotein mRNA vaccine was unequivocally positive in all individuals with a high level of RBD IgG antibodies and abundant generation of IFN gamma in response to S1 peptides. The individuals with a positive past history of COVID-19 were good responders in IgA class and frequently had N and S2 antibodies. Therefore, the presence of N and S2 antibodies may indicate previous SARS CoV2 exposure.

## Figures and Tables

**Figure 1 vaccines-09-00488-f001:**
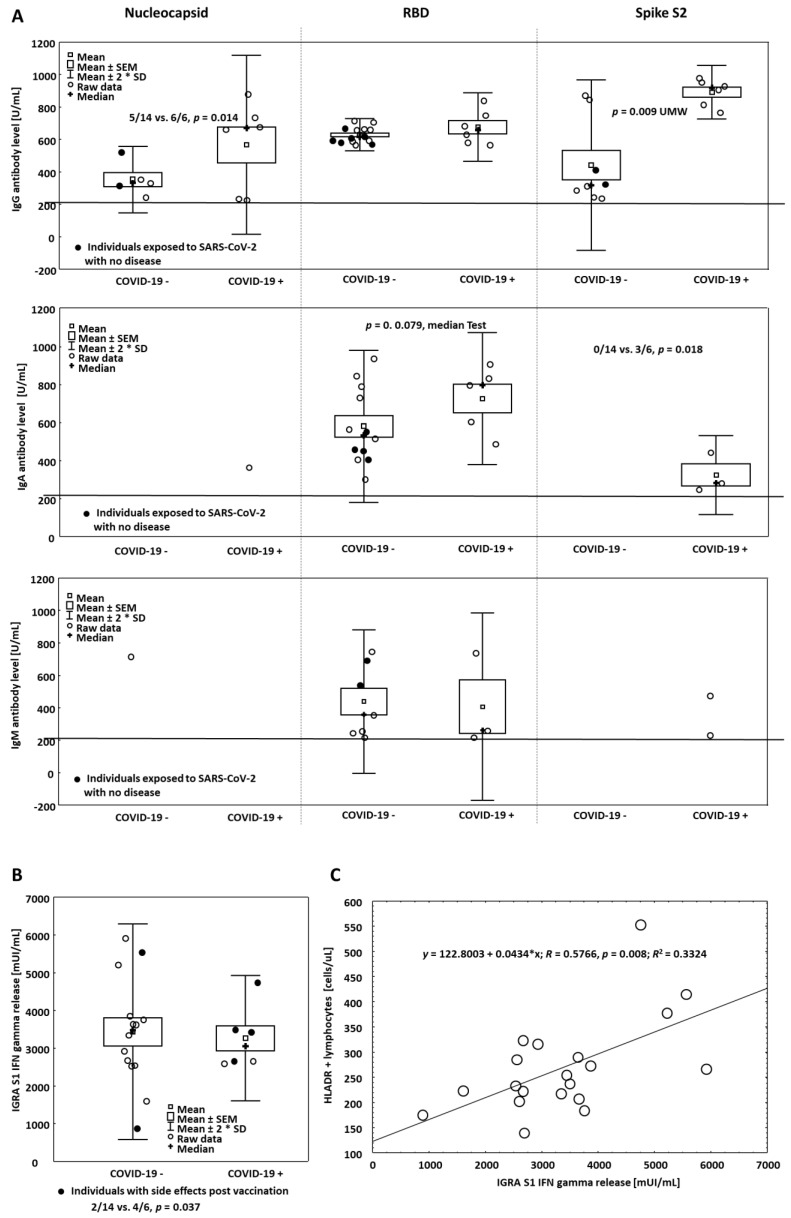
Panel (**A**) Levels of SARS-COV-2 antibodies after vaccination in individuals having and lacking COVID-19 past history. The horizontal line represents the threshold value of positivity (only the positive sera results are shown, metrics represent both the statistics of individual values and frequencies in the groups). Panel (**B**) IGRA results in the groups as above, closed circles indicate individuals with post-vaccination side effects. Note that the patients who had experienced the disease had side effects after vaccination more frequently than those not having COVID-19 in the past but both groups did not differ in IFN gamma response to S1 peptides. Panel (**C**) Correlation curve between IGRA results and numbers of HLADR + lymphocytes in the blood.

**Figure 2 vaccines-09-00488-f002:**
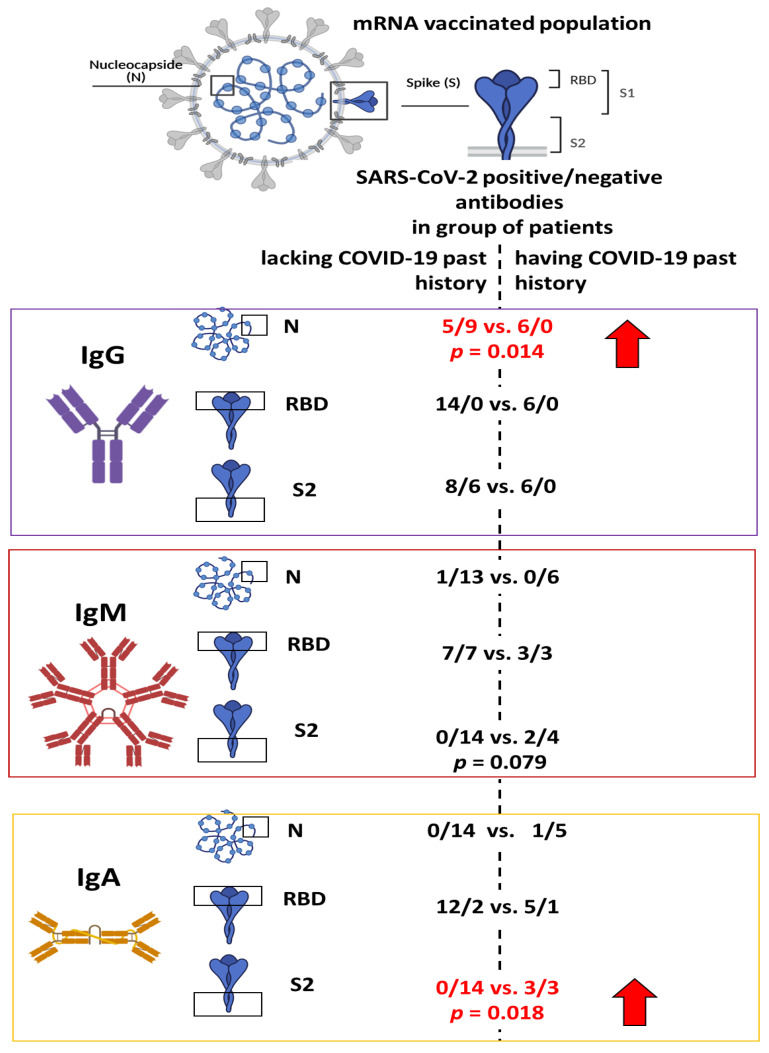
SARS-COV-2 antibodies in the individuals lacking or having COVID-19 prior to vaccination after completion of the immunization. Note that positive history of COVID-19 was associated with the presence of N and S2 antibodies.

**Figure 3 vaccines-09-00488-f003:**
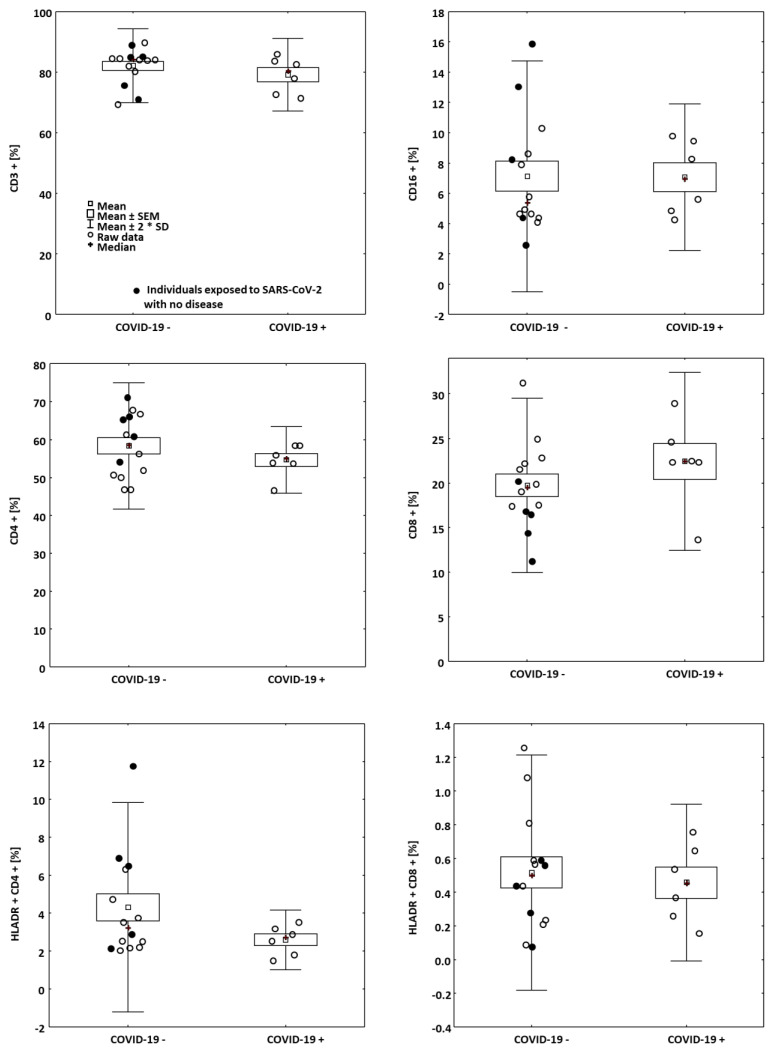
Proportions of lymphocyte subpopulation in the blood in the patients having and lacking COVID-19 in the past.

## Data Availability

All data are stored in the Hirszfeld Institute of Immunology and Experimental Therapy, Polish Academy of Sciences, in accordance with the institution’s data management plan.

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
