# Peer review of "Immune Response to COVID-19 mRNA Vaccine—A Pilot Study"

_vaccines, 2021, doi:10.3390/vaccines9050488_

Round 1

Reviewer 1 Report

The present Communication manuscript is well-structured, well-written and easy to understand.

The mRNA vaccine developed either by Pfizer BioNTech or Moderna made the vaccination roll out quite recently possible. Introduction of the vaccine was unprecedently fast, a consequence of outstanding scientific achievements.

The author observed that at 25 days after the second dose of the Pfizer BioNTech mRNA vaccine all immunized individuals developed anti-RBD antibodies at an average level of over 600 U/ml. The effect of the past history of COVID-19 was negligible in IgG class antibodies but was observed when IgA antibodies were evaluated. This may suggest that COVID-19 patients have a good memory of IgA class response to S1 SARS-CoV-2 peptides. Indeed, the recent study of Sterlin et al. documents prevalence of an IgA response at an early stage of COVID-19 with a high representation of IgA positive plasma blast at the mucosal barrier.

It also addresses a subject that is of great interest in the scientific community. A very large amount of work was involved in the study, the figure is informative and as far as I can determine, the work is solid.  The results are always new or interesting. 

I no any comments for the author. I strong suggest the editor accept it at the present version.

Author Response

Reviewer 1:
Thank you very much for your the most favorable opinion.

Reviewer 2 Report

The present communication paper deals with a pilot study about anti-SARS-CoV-2 vaccination-induced immune responses. Despite the fact that the topic is of high significance, the number of persons invited for the present study is small. A higher number should allow better statistics.

Paragraph 2.2: Authors should describe in detail the present assay. If S1 domain and the mitogen were provided by the kit or were purchased separately should be specified.

Paragraph 2.3: Please specify the method for isolating PBMCs and whether these cells were stimulated or not.

Results section:

Line 116: Is there a difference in the side effects between COVID-19+ and COVID-19- persons? Authors should incorporate a table with them  and discuss if there is a correlation among the severity of side effects and the magnitude of immune responses detected.

Lines 149-152: Authors have not provided the figure for IgM class antibodies. Also, there are two figure legends. Please correct appropriately.

Lines 161-169: Authors supported that there is a positive correlation between numbers of HLADR+ and IFNg release. On the other hand authors supported that there were no differences in " ...percentages and numbers of the studied subpopulations ...". These findings contradict each other since IFNg is a result of PBMCs stimulation with S1.

Major concerns: 

Have the authors tested vaccination-induced antibody response after the first dose? There are published findings that previously-exposed persons to SARS-CoV-2 were capable of eliciting higher immune responses than non-exposed ones.

Moreover, why they support that anti-S2 may indicate previous SARS-CoV-2 exposure, since S2 is domain of Spike protein? Only, anti-N antibody response can be considered a good marker for previous exposure to the virus. There is relative literature regarding this. 

Author Response

Reviewer 2:
We appreciate very much the reviewers comments which we used as it is listed below:
Comment 1. Paragraph 2.2: Authors should describe in detail the present assay. If S1 domain and the mitogen were provided by the kit or were purchased separately should be specified.
Response: In the revised version the following sentence was included:
page 3, line 100-101:
The tubes coated either with S1 domain or mitogen were provided by the manufacturer.

Comment 2. Paragraph 2.3: Please specify the method for isolating PBMCs and whether these cells were stimulated or not.
In the revised version the following explanation was included:
page 3, line 106-109:
Response: Staining was performed according to the lyse and stain approach using monoclinal antibodies: CD45 FITC (clone 2D1), CD4 PE/CF594 (clone RPA-T4, BD Biosciences, USA), CD3 PE-Cy7 (clone UCHT1), CD8 BV510 (clone SK1), CD16 PE (clone B73.1), HLADR BV786 (clone L243, BioLegend, San Diego, CA, USA) and the results read in the Fortessa flow cytometer (BD Biosciences, USA).

Comment 3. Line 116: Is there a difference in the side effects between COVID-19+ and COVID-19- persons? Authors should incorporate a table with them and discuss if there is a correlation among the severity of side effects and the magnitude of immune responses detected.
Response: Details of side effects were added to the text see lines: 119-124. In addition Figure 1 panel B shows IGRA results in individuals having or lacking post-vaccination side-effects (as it was shown also in the primary version), and in the revised version the following sentences were included (page 3, lines 123-130):
The side effects were seen from 2 to 7 (median:2) day after the second dose, they were rather mild but raised awareness. Symptoms included: muscle and joint pain, dyspnoea, weakness, fever, headache, eye pain. Only one person claimed the presence of moderate side-effects which included the lymph nodes enlargement under the arm and in the supraclavicular area what lasted about 10 days. The response to immunization was not different in cases having or lacking vaccination side-effects.

Comment 4. Lines 149-152: Authors have not provided the figure for IgM class antibodies. Also, there are two figure legends. Please correct appropriately.
Response: IgM antibodies levels have been included to the Figure 2 and finally the figure looks like in the revised manuscript.

In addition the following explanatory sentence was included in the text (page 4 line 153-157):
IgM response against RBD was rather higher in the individuals they received vaccines but denied the disease as compared to those who were vaccinated and previously experienced COVID-19 .It is our belief that it reflects a primary response to Pfizer BioNTech vaccine in individuals not contracting SARS-CoV2- before.

Comment 5. Lines 161-169: Authors supported that there is a positive correlation between numbers of HLADR+ and IFNg release. On the other hand, authors supported that there were no differences in " ...percentages and numbers of the studied subpopulations ...". These findings contradict each other since IFNg is a result of PBMCs stimulation with S1.
The latter comment was considered together with Reviewer 3 comment 6 which sounds:
Can you also include figures depicting percentage of T-cell populations between people with and without COVID-19 history?
Response: In the revised version we addressed both comments replacing the pertinent paragraph:
Page 4, line 176-182:
Between the individuals having or denying COVID-19 as well as having or lacking side effects no significant differences were found in the metrics of the readings (Figure 3). However, when the whole cohort was evaluated, independently on the presence or absence either COVID-19 positive history or side effects, the IGRA response (after S1 domain stimulation) was correlated with the numbers of HLADR+ T lymphocytes in the fresh blood (Figure 2B).

Comment 6. Have the authors tested vaccination-induced antibody response after the first dose? There are published findings that previously exposed persons to SARS-CoV-2 were capable of eliciting higher immune responses than non-exposed ones.
Moreover, why they support that anti-S2 may indicate previous SARS-CoV-2 exposure, since S2 is a domain of Spike protein? Only, anti-N antibody response can be considered a good marker for previous exposure to the virus. There is relative literature regarding this.
Unfortunately, no antibody response to the first dose was determined.
Response:  In page 8, line 211-217 the following phrase was removed:
The presence of N and S2 antibodies after anti-RBD vaccination in the persons with negative COVID-19 history needs some explanation. These antibodies might be primarily raised by previous common betacoronavirus exposure and then recalled by vaccination due to the presence in the sera of convalescents from betacoronavirus infections of a high (in COVID-19 cases) and rather low level (after possible common betacoronavirus infections), but still, of cross-reactivity between betacoronaviruses in IgG N and S antibodies [8,9]
and replaced by:
S2 antibodies levels measured after the vaccination was rather higher in the individuals having than in those lacking COVID-19 in the past. The similar association was not seen when anti- RBD antibody levels were evaluated (Figure 2A). In addition, all COVID-19 patients had S2 antibodies but not all denying the disease (Figure 1). It suggests the environmental exposure may play a role in shaping S2 antibody response to vaccination [8]. Likely including other viruses as S2 peptides are similar in structure among coronaviruses This observation warrants further study on a large cohort. The presence of N antibodies are strongly suggestive of the previous positive COVID-19 history (Figure 1, Figure 2).

Reviewer 3 Report

The manuscript titled "Immune response to COVID-19 mRNA vaccine - a pilot study" describes and compares immune response between people who have received Pfizer COVID-19 vaccine with or without a past history of COVID-19 exposure. The manuscript is clearly written, results appropriately interpreted and is of interest to the audience. The following points need to be addressed by the authors:

Major comments:

  1. Were any side effects noted in the people in the study between 1-5 days after vaccination?
  2. Under the Methods section, does the sentence "In the studied cohort 6 persons had a previous history of COVID-19 (in 4 to 5 months prior to vaccination), and 6 vaccinated persons had post-vaccination side effect symptoms." describe the same 6 people in the first and the second halves?
  3. Please include the clones for all the fluorescent antibodies used.
  4. Was FlowJo used for downstream analysis? Also describe the gating strategy in the methods. Did you also stain and analyze samples for IFNg by flow cytometry? 
  5. On page 6, under results, please edit: "S2 and N specific antibodies were seen in 2 out of ? and 1 out of ? cases, respectively, and were not associated with a past history of COVID-19."
  6. Can you also include figures depicting percentage of T-cell populations between people with and without COVID-19 history?
  7. Do you think number of people included in this study is sufficient? If not, please include this concern in the Discussion section.

Minor comments:

  1. Page 3, Results: change to: "The individuals who had experienced this disease had significantly more IgG antibodies against"
  2. Replace "persons" in the text with "people" or "subjects"

Author Response

Comment 1. Were any side effects noted in the people in the study between 1-5 days after vaccination?
Response: On page 3, lines 123-130 the following sentences were included:
The side effects were seen from 2 to 7 (median:2) days after the second dose, they were rather mild but raised awareness. Symptoms included: muscle and joint pain, dyspnoea, weakness, fever, headache, eye pain. Only one person claimed the presence of moderate side-effects which included the lymph nodes enlargement under the arm and in the supraclavicular area what lasted about 10 days. The response to immunization was not different in cases having or lacking vaccination side-effects.

Comment 2. Under the Methods section, does the sentence "In the studied cohort 6 persons had a previous history of COVID-19 (in 4 to 5 months prior to vaccination), and 6 vaccinated persons had post-vaccination side effect symptoms." describe the same 6 people in the first and the second halves?
Response: Sorry that it was not clearly written, the sentence in the revised version sounds as follows (page 2, lines 82-83):
 In the studied cohort, 6 people had a previous history of COVID-19 (in 4 to 5 months prior to vaccination), and 6 vaccinated people had post-vaccination side effect symptoms. Both groups overlapped as 4 individuals had both COVD-19 in the past and post-vaccination side effects.

Comment 3. Please include the clones for all the fluorescent antibodies used.
Comment 4. Was FlowJo used for downstream analysis? Also, describe the gating strategy in the methods. Did you also stain and analyze samples for IFNg by flow cytometry?
Response: All clones were described as suggested. In the method section the following sentence was added (page 3, lines 106-113)
Staining was performed according to the lyse and stain approach using monoclinal antibodies:CD45 FITC (clone 2D1), CD4 PE/CF594 (clone RPA-T4, BD Biosciences, USA), CD3 PE-Cy7 (clone UCHT1), CD8 BV510 (clone SK1), CD16 PE (clone B73.1), HLADR BV786 (clone L243, BioLegend, San Diego, CA, USA) and the results read in the Fortessa flow cytometer (BD Biosciences, USA). Living cells (stained with LIVE/DEAD Fixable Dead Cell Stain Kit with BV421 fluorochrome, Life Technologies Carlsbad, CA, USA) were evaluated. The gating was done using both CD45 and side scatter signals. NovoExpress Software (Agilent Santa Clara, CA, the USA) was used for subpopulation analysis.
Unfortunately, we did not stain lymphocytes for IFN gamma intracellularly.

Comment 5. On page 6, under results, please edit: "S2 and N specific antibodies were seen in 2 out of ? and 1 out of ? cases, respectively, and were not associated with a past history of COVID-19."
Response: In the revised version the sentence sounds as follows (page 4, lines 153-157):
IgM S2 and N antibodies were seen in 2 and 1 case out of 20 individuals examined, respectively. Their presence was not associated with the past history of COVID-19. IgM response against RBD was rather higher in the individuals they received vaccines but denied the disease as compared to those who were vaccinated and previously experienced COVID-19 .It is our belief that it reflects a primary response to Pfizer BioNTech vaccine in individuals not contracting SARS-CoV2- before.

Comment 6. Can you also include figures depicting the percentage of T-cell populations between people with and without COVID-19 history?
Response: The relevant figure was added in the revised version (Figure 3):

Figure 3. Proportions of lymphocyte subpopulation in the blood in the patients having and lacking COVID-19 in the past.

Comment 7. Do you think number of people included in this study is sufficient? If not, please include this concern in the Discussion section.
Response: Yes, we are aware of constraints due to rather small number of individuals investigated and it is expressed in the discussion section page 8, lines 229-230:
The limitation of this study is due to rather a small number of people investigated, but the results show the area of interest for planning large scale investigations.

Minor comments:
1. Page 3, Results: change to: "The individuals who had experienced this disease had significantly more IgG antibodies against"

The quoted sentence was modified as requested.

2. Replace "persons" in the text with "people" or "subjects"

Round 2

Reviewer 2 Report

The authors answered satisfactorily in most of the questions. Thus, the manuscript has been significanlty improved.

However, they did not give details regarding PBMC isolation method and if they used fresh or frozen cells (Lines 106-109, Materials and Methods section).

Author Response

I appreciate very much the critical comments which I followed while revising the manuscript. In addition to some spelling and grammar improvements the explanation (sorry for overlooking this detail) you requested was added as follows:

p3 line 105: 

Staining of unfractionated fresh blood was performed according to the lyse and stain approach...